# O group is a protective factor for COVID19 in Basque population

**Maider Muñoz-Culla[1], Andres Roncancio-Clavijo[2], Bruno Martínez[3], Miriam Gorostidi-Aicua[1], Luis Piñeiro[4], Arkaitz Azkune[5], Ainhoa Alberro[1], Jorge Monge-Ruiz[6], Tamara Castillo-Trivino[1,2,3,4,5,6,7], Alvaro Prada[2], David Otaegui[1]***

1 Biodonostia Health Research Institute, Group of Multiple Sclerosis, San Sebastián, Spain, 2 Immunology Department, Donostia University Hospital, Osakidetza Basque Health Service, San Sebastián, Spain, 3 UGC Laboratories Gipuzkoa, Osakidetza Basque Health Service, San Sebastián, Spain, 4 Microbiology Department, Donostia University Hospital, Osakidetza Basque Health Service, San Sebastián, Spain, 5 Infectious Disease Department, Donostia University Hospital, Osakidetza Basque Health Service, San Sebastián, Spain, 6 Basque Center for Blood Transfusion and Human Tissues, Osakidetza Basque Health Service, Galdakao, Spain, 7 Neurology Department, Donostia University Hospital, Osakidetza Basque Health Service, San Sebastián, Spain

* david.otaegui@biodonostia.org

**Data Availability Statement:** All relevant data are within the manuscript.

**Funding:** This project has been supported by Instituto de Salud Carlos III (COV20/00314) that

## Abstract

ABO blood groups have recently been related to COVID19 infection. In the present work, we performed this analysis using data from 412 COVID19 patients and 17796 blood donors, all of them from Gipuzkoa, a region in Northern Spain. The results obtained confirmed this relation, in addition to showing a clear importance of group O as a protective factor in COVID19 disease, with an OR = 0.59 (CI95% 0.481–0.7177, p<0.0001) while A, B and AB are risk factors. ABO blood groups are slightly differently distributed in the populations and therefore these results should be replicated in the specific areas with a proper control population.

## Introduction

COVID19 is a pandemic disease caused by Severe Acute Respiratory Syndrome Coronavirus 2 (SARS-CoV-2). This disease has rapidly become the most important world health challenge in the last century.

Despite an incredible and collaborative research effort in the last months, the pathogenesis and the clinical symptoms of COVID19 are poorly understood. Several works have been published to understand the great heterogeneity in the infection and the clinical manifestations of SARS-CoV-2 in different patients. Several biomarkers have been proposed as risk or protective factors. In this scenario, a recent paper, demonstrating the importance of collaborative networks, presents the results of a Genome-wide analysis focused on COVID19 patients [1]. One of the associated loci in chromosome 9, point to an association between ABO blood group and risk of infection by SARS-CoV-2. This association with the ABO blood group has been previously reported in other series showing that Group A is a risk factor while Group O seems to be a protective factor against SARS-CoV-2 infection.

includes European Support (FEDER). There was no additional external funding received for this study. The funders had no role in study design, data collection and analysis, decision to publish, or preparation of the manuscript.

**Competing interests:** The authors have declared that no competing interests exist.

ABO groups are distributed differently based on ethnic origin and, therefore, the results from these studies are highly dependent on the distribution followed by the control group in each region. With this in mind, the main aim of this short report is to study the ABO group distribution in COVID19 patients in the region of Gipuzkoa (Basque country).

## Methods

Blood groups from COVID19 patients were anonymously retrieved from electronic clinical history. All the included patients were symptomatic and had tested positive for SARS-CoV-2 infection by qPCR technique at the Microbiology Department of Donostia University Hospital. The kits used for the qPCR came from different suppliers and all of them are approved for diagnostic purposes.

The distribution of the ABO groups from COVID19 patients was compared to that from the general Basque population, obtained from the Basque Blood Bank database and preserving the anonymity of all the participants. Both the anonymous data and the realization of the project have been approved by "the OSI Donostialdea Ethical Committee in Clinical Research" (Code: API-GRA-2020-01). Besides, the used information and methodology are in accordance with the adequate guidelines and regulations.

When analyzing the data, SPSS software was used (v 20) and for the analysis of the distribution chi-square test was applied.

Patients were divided by groups based on severity, following the next criteria: they were characterized as Mild cases when clinical symptoms were moderate with no signs of pneumonia on imaging. Moderate cases, instead, were those individuals that manifested fever and respiratory symptoms and showed radiological findings of pneumonia. Finally, Severe cases were those meeting any of the following criteria: (1) Respiratory distress ($\geqq$30 breaths/ min); (2) Oxygen saturation$\leqq$93% at rest; (3) Arterial partial pressure of oxygen (PaO2)/ fraction of inspired oxygen (FiO2)$\leqq$300mmHg (l mmHg = 0.133kPa). This classification was based on the one proposed by the Chinese National Health Commission on March 3, 2020 [2].

## Results

Our cohort includes 412 COVID19 patients and 17796 anonymous blood donors from the same geographical area (Gipuzkoa). The average age of COVID19 patients is 57.64 years, with a 68.45% of women. Referring to severity, 49.29% of the patients were characterized as moderate, 8.92% as mild and 41.78% as severe.

The distribution of the ABO groups is shown in Table 1. In line with the published works, our results also show a higher frequency of group A in COVID19 patients when compared to control group, in this case Basque population, (48.3% vs 40.65%, p = 0.00179), while group O is less represented (39.08 vs 52.19%, p<0.0001). The group A presents an OR of 1.36 (CI95% 1.12–1.66, p = 0.0019), while O group's OR is 0.5876 (CI95% 0.481–0.7177, p<0.0001). Group B and AB also show a significantly different distribution (p = 0.00186 and p = 0.012204) with an OR = 1.76 (CI95% 1.24–2.49) and OR = 1.98 (CI95% 1.19–3.31), respectively. Nonetheless, since those groups are less frequent in the population, these results should be taken with caution.

Furthermore, if ABO group is analyzed as a dichotomous variable; defined as having any antigen (A,B and AB) or none (O); the results highlight the protective role of group O, being the OR for no-O equal to 1.7019 (CI95% 1.94–2.08, p<0.0001). No difference has been found in ABO group distribution with severity.

**Table 1. ABO group distribution in the Gipuzkoa cohort.**

|  | A | B | AB | O | TOTAL |
|---|---|---|---|---|---|
| **Blood donors** | **40.65% (7235)** | **5.16% (918)** | **1.20% (355)** | **52.19% (9288)** | **17796** |
| **COVID19 patients** | **48.30% (199)** | **8.74% (36)** | **3.88% (16)** | **39.08% (161)** | **412** |
| p-value | 0.00179 | 0.00186 | 0.012204 | <0.0001 | |
| **COVID19 patients** | **A** | **B** | **AB** | **O** | **TOTAL** |
| Moderate | 48.57%(51) | 50.00% (8) | 45.54% (5) | 50.62%(41) | **105** |
| Light | 10.48%(11) | 6.25%(1) | 0.00%(0) | 8.64%(7) | 89 |
| Severe | 40.95%(43) | 43.75%(7) | 54.55%(6) | 40.74%(33) | 19 |
| p-value | n.s. | | | | 213 |

P-values from the Chi-square test are shown for the comparison of each group distribution in blood donors vs COVID19 patients. ns = No significant (p value >0.05).

## Discussion

Our data shows the importance of the absence of immune antigens defined by group O, as a protective factor for Sars-CoV-2 virus infection. These results are consistent with the pre-printed works conducted in the Chinese population [3] and in the New York cohort [4], and also with the genotyping studies performed in the Italian and Spanish population (including samples from Gipuzkoa). Due to the different distribution of the ABO groups by ethnicity, the validation of these kind of observations, in each region and with the proper controls, is important to achieve a better understanding of how the virus is spreading in the population. In our study, the group O distribution differs from that of other series, seeming more protective in the population from Gipuzkoa. In Fig 1 we represent the differences between the populations in the two more represented blood groups (O and A).

The observed different distribution seems to be related to the infection event and not to the course of the disease. The mechanisms behind these observations remain unknown. It has been proposed that the presence of anti-A Antibodies could be protective against viral entry into lung epithelium or that it may be the fact that O group presents an altered glycosiltransferase activity and, therefore, an increased clearance of Von Willebrand factor, that could protect O group patients from the COVID19- related microvascular thrombosis and endothelial dysfunction [5].

In conclusion, our data confirms the idea that O group is a protective factor for COVID19, and the presence of any antigen (A, B and AB) are a risk factor to suffer the disease. Our results

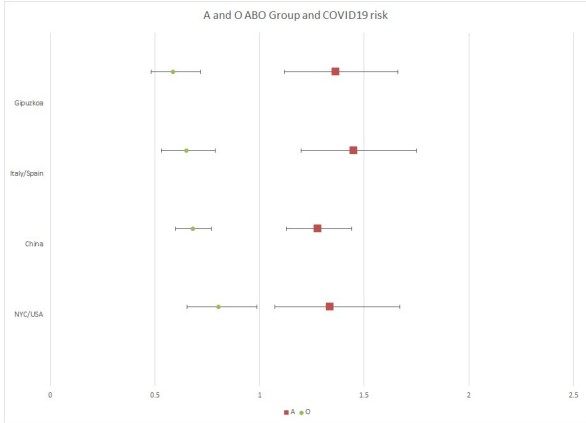

**Fig 1. Odd ratio of A (red square) and O group (green circle) from our study and the ones found in the literature.**

also support the need for further studies in each region with proper population controls. The biological implications of these observations deserve future investigations to shed light on the mechanisms behind COVID19 infection risk.

## Acknowledgments

Authors want to thank all the clinicians that have worked so hard these last months and to all the COVID19 patients.

## Author Contributions

**Conceptualization:** Andres Roncancio-Clavijo, David Otaegui.

**Data curation:** Maider Muñoz-Culla, Bruno Martínez, Arkaitz Azkune, Jorge Monge-Ruiz.

**Formal analysis:** Miriam Gorostidi-Aicua, Ainhoa Alberro, David Otaegui.

**Investigation:** Luis Piñeiro, Arkaitz Azkune, Ainhoa Alberro, Tamara Castillo-Trivino, David Otaegui.

**Resources:** Tamara Castillo-Trivino, Alvaro Prada, David Otaegui.

**Supervision:** Alvaro Prada.

**Writing – original draft:** Ainhoa Alberro, David Otaegui.

**Writing – review & editing:** Maider Muñoz-Culla, Andres Roncancio-Clavijo, Luis Piñeiro, Tamara Castillo-Trivino, David Otaegui.

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
