## [Decision Letter · Decision Letter 0]

4 Dec 2020

PONE-D-20-36036

O Group is a protective factor for COVID19 in Basque population

PLOS ONE

Dear Dr. Otaegui,

Thank you for submitting your manuscript to PLOS ONE. After careful consideration, we feel that it has merit but does not fully meet PLOS ONE’s publication criteria as it currently stands. Therefore, we invite you to submit a revised version of the manuscript that addresses the points raised during the review process.

The article is interesting but it needs some extra work before it can be accepted.

We look forward to receiving your revised manuscript.

Kind regards,

Prof. Raffaele Serra, M.D., Ph.D

Academic Editor

PLOS ONE

Journal Requirements:

2.Please provide additional details regarding participant consent. In the ethics statement in the Methods and online submission information, please ensure that you have specified (1) whether consent was informed and (2) what type you obtained (for instance, written or verbal, and if verbal, how it was documented and witnessed). If your study included minors, state whether you obtained consent from parents or guardians. If the need for consent was waived by the ethics committee, please include this information.

3. To comply with PLOS ONE submission guidelines, in your Methods section, please provide additional information regarding your statistical analyses and ensure you have included (1) the name and version of any software package used, alongside any relevant references, (2) the technical details or procedures required to reproduce the analysis . For more information on PLOS ONE's expectations for statistical reporting, please see https://journals.plos.org/plosone/s/submission-guidelines.#loc-statistical-reporting.

4.Thank you for stating in your Funding Statement:

 [This project has been partially supported by Instituto de Salud Carlos III (COV20/00314).]. 

Additional Editor Comments (if provided):

The article is potentially interesting but some concerns need to be addressed.

Reviewers' comments:

Reviewer's Responses to Questions

**Comments to the Author**

1. Is the manuscript technically sound, and do the data support the conclusions?

Reviewer #1: Yes

2. Has the statistical analysis been performed appropriately and rigorously? 

Reviewer #1: Yes

3. Have the authors made all data underlying the findings in their manuscript fully available?

Reviewer #1: Yes

4. Is the manuscript presented in an intelligible fashion and written in standard English?

Reviewer #1: Yes

5. Review Comments to the Author

Reviewer #1: In this study, the authors compare the distributions of blood group types in (apparently) symptomatic COVID19 patients diagnosed in Gipuzkoa (Basque country, Spain) with the distributions of blood group types found in the general population, as inferred from a local Basque blood bank. They conclude that O blood group is protective, whereas A blood group is a risk factor. This study joins a handful of others, from various parts of the world, confirming an influence of blood group on vulnerability to COVID19 infection.

I have a few suggestions for the authors to consider:

1. In the abstract, I would suggest moving what is currently the second sentence (the call for replication of the work in different regions of the world) to the end of the abstract text, after the main results have been presented. I would also encourage the authors to give, in the abstract, some quantification of the degree of protection conferred by group O blood (e.g, the odds ratio).

2. The authors state that all patients had tested positive for SARS-CoV-2 infection at Donostia University Hospital “according to the actual guidelines.” Could they please describe briefly those guidelines? Were these patients symptomatic, asymptomatic, or a mix of the two? Later, in the results, it appears that all the COVID patients had some level of clinical symptoms, but it would be good to clarify that asymptomatic individuals were excluded from the study, if that is indeed the case.

3. The results section states “According with the literature, in our data group A is more frequent in COVID19 patients (48.3% . . . )”; the authors need to be explicit that they are comparing this with the frequency of group A in the general Basque population.

4. I think the authors could add value to their paper by testing whether blood groups A, B, and AB differ from each other in terms of susceptibility to COVID infection. The authors emphasize group A, but it seems like group B might be still more susceptible, and AB perhaps the most susceptible.

5. The authors state that “No difference has been found in ABO group distribution with severity”, but the readers cannot evaluate this claim, because the data are not shown. I would encourage the authors to include the data broken down into severity categories, along with some formal statistical analysis of disease severity effects. This could be done in an on-line electronic supplement if the authors don’t wish to include it in the main text of the manuscript.

6. The manuscript needs a final light editing to correct numerous small problems with English grammar. (I would have called some of these out, but my copy of the manuscript does not have line numbers.)

6. PLOS authors have the option to publish the peer review history of their article (what does this mean?). If published, this will include your full peer review and any attached files.

Reviewer #1: No

---

## [Author Response · Author response to Decision Letter 0]

26 Jan 2021

POINT-BY-POINT RESPONSE TO THE REVIEWERS

Reviewer #1

In this study, the authors compare the distributions of blood group types in (apparently) symptomatic COVID19 patients diagnosed in Gipuzkoa (Basque country, Spain) with the distributions of blood group types found in the general population, as inferred from a local Basque blood bank. They conclude that O blood group is protective, whereas A blood group is a risk factor. This study joins a handful of others, from various parts of the world, confirming an influence of blood group on vulnerability to COVID19 infection.

I have a few suggestions for the authors to consider:

1. In the abstract, I would suggest moving what is currently the second sentence (the call for replication of the work in different regions of the world) to the end of the abstract text, after the main results have been presented. I would also encourage the authors to give, in the abstract, some quantification of the degree of protection conferred by group O blood (e.g, the odds ratio).

We thank to the reviewer for this comment. As suggested we change the abstract and add the required information.

2. The authors state that all patients had tested positive for SARS-CoV-2 infection at Donostia University Hospital “according to the actual guidelines.” Could they please describe briefly those guidelines? Were these patients symptomatic, asymptomatic, or a mix of the two? Later, in the results, it appears that all the COVID patients had some level of clinical symptoms, but it would be good to clarify that asymptomatic individuals were excluded from the study, if that is indeed the case.

Thanks for the comment. We added this information to the method section of the manuscript to clarify that all the COVID19 patients included in the study were symptomatic and how the positivity for the virus was tested. Approved SARS_CoV-2 qPCR test has been performed for all the samples at Microbiology department in the Donostia University Hospital. The moment in which all data for this study were retrieved, was a moment in the pandemic in which all the patients that came to the Hospital were Symptomatic.

3. The results section states “According with the literature, in our data group A is more frequent in COVID19 patients (48.3% . . . )”; the authors need to be explicit that they are comparing this with the frequency of group A in the general Basque population.

Thanks for notice it. We have added this to the manuscript and have rewritten the sentence to avoid any misunderstanding.

4. I think the authors could add value to their paper by testing whether blood groups A, B, and AB differ from each other in terms of susceptibility to COVID infection. The authors emphasize group A, but it seems like group B might be still more susceptible, and AB perhaps the most susceptible.

We change our results section adding tests for each group.

5. The authors state that “No difference has been found in ABO group distribution with severity”, but the readers cannot evaluate this claim, because the data are not shown. I would encourage the authors to include the data broken down into severity categories, along with some formal statistical analysis of disease severity effects. This could be done in an on-line electronic supplement if the authors don’t wish to include it in the main text of the manuscript.

We add in Table 1 a distribution of the ABO groups with severity.

6. The manuscript needs a final light editing to correct numerous small problems with English grammar. (I would have called some of these out, but my copy of the manuscript does not have line numbers.)

Thanks to the reviewer for the comment. We have reviewed the manuscript and polish the language, so we hope that now the English grammar is correct.

---

## [Decision Letter · Decision Letter 1]

9 Feb 2021

PONE-D-20-36036R1

O Group is a protective factor for COVID19 in Basque population

PLOS ONE

Dear Dr. Otaegui,

Thank you for submitting your manuscript to PLOS ONE. After careful consideration, we feel that it has merit but does not fully meet PLOS ONE’s publication criteria as it currently stands. Therefore, we invite you to submit a revised version of the manuscript that addresses the points raised during the review process.

The manuscript was improved but it needs further revision as suggested by the reviewer.

We look forward to receiving your revised manuscript.

Kind regards,

Prof. Raffaele Serra, M.D., Ph.D

Academic Editor

PLOS ONE

Additional Editor Comments (if provided):

While the manuscript was improved it needs further reworking.

Reviewers' comments:

Reviewer's Responses to Questions

**Comments to the Author**

1. If the authors have adequately addressed your comments raised in a previous round of review and you feel that this manuscript is now acceptable for publication, you may indicate that here to bypass the “Comments to the Author” section, enter your conflict of interest statement in the “Confidential to Editor” section, and submit your "Accept" recommendation.

Reviewer #1: (No Response)

2. Is the manuscript technically sound, and do the data support the conclusions?

Reviewer #1: Partly

3. Has the statistical analysis been performed appropriately and rigorously? 

Reviewer #1: Yes

4. Have the authors made all data underlying the findings in their manuscript fully available?

Reviewer #1: Yes

5. Is the manuscript presented in an intelligible fashion and written in standard English?

Reviewer #1: No

6. Review Comments to the Author

Reviewer #1: I reviewed this manuscript previously, so my current reading emphasized looking for changes since the first draft. The reviewers successfully implemented many changes, and I think this revision is improved.

I do think, however, that given that the new analyses shown in Table 1 now show that blood types B and AB are even greater risk factors than blood type A, the rest of the discussion in the manuscript (which emphasizes effects of blood group A and ignores B and AB) seems now incomplete and not quite appropriate. I would encourage the authors to modify the discussion to reflect their results: O is protective, and A and especially B are risk factors.

The authors also mention in passing that there seems to be a higher risk of infection for men. If they wish to make this point, I think they should support it with some sort of quantitative statistical test. Given that men and women may have different exposures to infection or different likelihoods of seeking medical care when sick, I don’t know how strong the evidence in Table 1 really is with regards to effects of sex. I’d encourage the authors to make a decision to either (1) choose to introduce the question of differential vulnerability to infection, and then formally test it, or (2) omit the issue altogether, and instead focus only on blood group. (Option 2 might be the best, but it’s really up to the authors.)

The manuscript still does not have line numbers, making it very difficult for me to suggest fixes to small grammatical errors. The paper still needs a final editing to correct numerous small problems.

7. PLOS authors have the option to publish the peer review history of their article (what does this mean?). If published, this will include your full peer review and any attached files.

Reviewer #1: No

---

## [Author Response · Author response to Decision Letter 1]

2 Mar 2021

POINT-BY-POINT RESPONSE TO THE REVIEWER

Reviewer #1

Reviewer #1: I reviewed this manuscript previously, so my current reading emphasized looking for changes since the first draft. The reviewers successfully implemented many changes, and I think this revision is improved.

We want to thanks the reviewer by her/his help to improve our manuscript.

I do think, however, that given that the new analyses shown in Table 1 now show that blood types B and AB are even greater risk factors than blood type A, the rest of the discussion in the manuscript (which emphasizes effects of blood group A and ignores B and AB) seems now incomplete and not quite appropriate. I would encourage the authors to modify the discussion to reflect their results: O is protective, and A and especially B are risk factors.

We agree with the reviewer that blood type B and even AB seems to be risk factor, however me try to focus our discussion in A and O group that are the two more represented in our patient’s population. Blood type B is only present in 36 patients and AB in 16 and that why we focus in the other 2 blood groups. We change our discussion to include the observation about B and AB group.

The authors also mention in passing that there seems to be a higher risk of infection for men. If they wish to make this point, I think they should support it with some sort of quantitative statistical test. Given that men and women may have different exposures to infection or different likelihoods of seeking medical care when sick, I don’t know how strong the evidence in Table 1 really is with regards to effects of sex. I’d encourage the authors to make a decision to either (1) choose to introduce the question of differential vulnerability to infection, and then formally test it, or (2) omit the issue altogether, and instead focus only on blood group. (Option 2 might be the best, but it’s really up to the authors.)

Thanks to the reviewer for this comment. We thought that could be interesting for the readers to have the data by gender but we are totally agree with the reviewer that with our experimental design we cannot say that men shows a higher infection and may be the data can reflect the effect of other issues. Therefore, to focus in the point that we think is more interesting, we rewrite the paragraph and the Table 1 to show just the data from patients vs controls.

The manuscript still does not have line numbers, making it very difficult for me to suggest fixes to small grammatical errors. The paper still needs a final editing to correct numerous small problems.

The manuscript has been reviewed by different persons to correct the grammatical problems and do it more readable. Line numbers had been included.

---

## [Decision Letter · Decision Letter 2]

19 Mar 2021

O Group is a protective factor for COVID19 in Basque population

PONE-D-20-36036R2

Dear Dr. Otaegui,

We’re pleased to inform you that your manuscript has been judged scientifically suitable for publication and will be formally accepted for publication once it meets all outstanding technical requirements.

Kind regards,

Prof. Raffaele Serra, M.D., Ph.D

Academic Editor

PLOS ONE

Additional Editor Comments (optional):

amended manuscript is acceptable

Reviewers' comments:

Reviewer's Responses to Questions

**Comments to the Author**

1. If the authors have adequately addressed your comments raised in a previous round of review and you feel that this manuscript is now acceptable for publication, you may indicate that here to bypass the “Comments to the Author” section, enter your conflict of interest statement in the “Confidential to Editor” section, and submit your "Accept" recommendation.

Reviewer #1: All comments have been addressed

2. Is the manuscript technically sound, and do the data support the conclusions?

Reviewer #1: Yes

3. Has the statistical analysis been performed appropriately and rigorously? 

Reviewer #1: Yes

4. Have the authors made all data underlying the findings in their manuscript fully available?

Reviewer #1: Yes

5. Is the manuscript presented in an intelligible fashion and written in standard English?

Reviewer #1: Yes

6. Review Comments to the Author

Reviewer #1: I think the revision is very successful in providing a more balanced discussion of the risk associated with A, B, and AB blood groups.

7. PLOS authors have the option to publish the peer review history of their article (what does this mean?). If published, this will include your full peer review and any attached files.

Reviewer #1: No

---

## [Editor Report · Acceptance letter]

29 Mar 2021

PONE-D-20-36036R2 

O Group is a protective factor for COVID19 in Basque population 

Dear Dr. Otaegui:

I'm pleased to inform you that your manuscript has been deemed suitable for publication in PLOS ONE. Congratulations! Your manuscript is now with our production department. 

Kind regards, 

on behalf of

Prof. Raffaele Serra 

Academic Editor

PLOS ONE